# Solid-State Fermentation of Soybean Meal with Edible Mushroom Mycelium to Improve Its Nutritional, Antioxidant Capacities and Physicochemical Properties

Jian Wang [1,2,†], Quanjin Jiang [1,†], Zhenyu Huang [1], Yan Wang [1,2], Hynek Roubik [3], Kai Yang [1,2,*], Ming Cai [1,2,*] and Peilong Sun [1,2,*]

1 Department of Food Science and Technology, Zhejiang University of Technology, Hangzhou 310014, China; wangjian1926@gmail.com (J.W.); jiangquanjin163@163.com (Q.J.); huangzy1120@163.com (Z.H.); wangyan062006@zjut.edu.cn (Y.W.)
2 Key Laboratory of Food Macromolecular Resources Processing Technology Research, Zhejiang University of Technology, China National Light Industry, Hangzhou 310014, China
3 Department of Sustainable Technologies, Faculty of Tropical AgriSciences, Czech University of Life Sciences Prague, 16500 Prague, Czech Republic; roubik@ftz.czu.cz
* Correspondence: yangkai@zjut.edu.cn (K.Y.); caiming@zjut.edu.cn (M.C.); sun_pl@zjut.edu.cn (P.S.); Tel.: +86-0571-88813778 (K.Y.)
† These authors contributed equally to this work.

**Abstract:** Soybean meal is a class of by-products obtained from the processing of soybean products. Despite its high nutritional value, the presence of glycoside isoflavones limits human use of soybean meal. This study evaluated the effect of solid-state fermentation (SSF) with different edible mushroom mycelia (*Pleurotus ostreatus*, *Hericium erinaceus*, and *Flammulina velutipes*) on the proximate composition, antioxidant properties, and physicochemical properties of fermented soybean meal powder (SP). The results revealed that fermented SP had a higher nutritional value when compared to SP. *P. ostreatus* was the most pronounced among the three species. Crude protein content was found to have increased by 9.49%, while the concentration of glutamate and aspartic acid increased by 23.39% and 23.16%, respectively. SSF process significantly increased the total polyphenol content (TPC) and aglycone isoflavone content by 235.9% and 324.12%, respectively, resulting in increased antioxidant activity (evaluated by the DPPH, •OH, ABTS$^+$ assays). Microstructural changes in fermented SP and nutrient degradation and utilization were observed. Thus, fermented SP can be used as a raw material with enhanced nutritional properties to develop new functional foods, such as plant-based foods represented by plant meat. It provides a promising approach for increasing the added value of soybean meal.

**Keywords:** soybean meal; solid-state fermentation; edible mushrooms; nutritional value

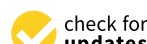



## 1. Introduction

As the global population grows and the climate changes, animal-derived foods will not be able to meet the increasing demand for human consumption. Therefore, there is an urgent need to find alternative sources of high-quality protein. Soybean meal is a by-product obtained from the pressing of soybean products into oil and has a rich nutritional value, with a high crude protein content of about 40–49% [1]. In addition, it is rich in trace elements such as calcium, iron, and phosphorus, as well as many vitamins, and has high bioaccessibility [2,3]. Soybean meal also contains isoflavones, a bioactive component with various beneficial effects, including the potential to reduce the risk of oxidative damage, menopausal symptoms, cancer, and osteoporosis [4,5]. These effects are primarily dependent on the content and structure of isoflavones [3]. Isoflavones in soybean meal typically exist in two forms: non-conjugated aglycones (daidzein, genistein, and glycitein) and conjugated glycosides (daidzin, genistin, and glycitin) [6]. Most of the isoflavones in

soybean meal are present in glycoside form [5]. The glycosides' high molecular weight and hydrophilicity result in lower bioavailability for this type of isoflavone, limiting absorption by the human body [6].

Fermentation is considered an effective method to enhance the nutritional value of soybean products [7]. Studies have shown that fermentation can degrade anti-nutritional compounds, such as phytates, lectins, and tannins, and improve the bioavailability of substrates such as aglycone-form isoflavones and free amino acids [8,9]. Solid-state fermentation (SSF) is a process in which microorganisms ferment solid substrates in a low-moisture environment. Compared to submerged fermentation (SmF), SSF has several advantages, such as lower water, aseptic, energy requirements, and higher volumetric productivity [10]. The fermentation process results in the production of various enzymes, such as proteases and amylases. These enzymes degrade carbohydrates and proteins into simpler compounds, thus altering the sensory and functional properties of soy products [7,11].

Edible mushrooms are a broad category of fungi that can be consumed by humans. Approximately 35 edible mushroom species are commercially grown, while almost 200 wild species are used for health applications [12,13]. The *Pleurotus ostreatus*, *Hericium erinaceus*, and *Flammulina velutipes* are among the species cultivated and produced worldwide and cataloged as Generally Recognized as Safe (GRAS) [14,15]. In recent years, research on SSF using edible mushrooms has gained traction, with more researchers investigating the potential value of fermented powder obtained from these edible mushrooms. For example, Espinosa-Páez et al. [16] evaluated the effect of SSF on *Phaseolus vulgaris* and *Avena sativa* with *P. ostreatus*, and observed increased antioxidant activity and total polyphenol content (TPC). Additionally, when in vitro digestion was simulated, fermented cereal and legume flours showed higher protein digestibility. In the other study, Suruga et al. [4] investigated the antioxidant activity and isoflavone species in fermented soybeans by SSF with *H. erinaceus* and found that fermentation improved the antioxidant capacity of soybeans, increased the content of isoflavone glycosides, and improved bioactivities. Fermented soybean curd residue with *Flammulina velutipes* was reported to produce polysaccharides, and the fermentation results showed strong DPPH radical scavenging activity [17]. The fermented soybean meal powders have the potential to serve as a functional ingredient in the food.

As reviewed by Mukherjee et al. [18], previous studies have evaluated the SSF of soybean meal with a single edible fungus. However, to date, few studies have compared the effects of different edible mushrooms on the SSF of soybean meal. So, the purpose of this study was to focus on three edible mushroom species, *Pleurotus ostreatus*, *Hericium erinaceus*, and *Flammulina velutipes*, and to characterize the nutritional value, antioxidant capacity, and physicochemical properties of the soybean meal fermented with these edible mushroom mycelia. This research reveals the potential of soybean processing by-products as functional ingredients for producing nutritious human food.

## 2. Materials and Methods

### 2.1. Materials

Soybean meal was purchased from Linyi Shansong Biological Products Co., Ltd. (Linyi, China). *Pleurotus ostreatus* ACCC 50476, *Flammulina velutipes* ACCC 51540, and *Hericium erinaceus* ACCC 50268 were acquired from China Agricultural Microbial Strain Storage Management Center (Beijing, China). Hydrochloric acid (HCl, 12 M), sodium hydroxide (NaOH, 96%), ethanol ($CH_2OH$, 95%), methanol ($CH_3OH$, 99.9%), phenol ($C_6H_6O$, 99.5%), sodium citrate ($C_6H_5O_7Na_3$, 98%), hydrogen peroxide ($H_2O_2$, 30%), magnesium sulfate ($MgSO_4 \bullet 7H_2O$, 99%), and sulfuric acid ($H_2SO_4$, 99.7%) were supplied by Shanghai Aladdin Ltd. (Shanghai, China). Glucose, peptone, potassium phosphate monobasic ($KH_2PO_4$, 99.5%), ferrous sulfate ($FeSO_4$, 90%), and vitamin $B_1$ were purchased from Shanghai Macklin Biochemical Technology Co., Ltd. (Shanghai, China). Gallic acid, 1,1-diphenyl-2-trinitrophenylhydrazine (DPPH), and 2,2′-biazo-bis-3-ethylbenzothiazoline-6-sulfonic acid ($ABTS^+$) were obtained from Beijing Solarbio Science and Technology Co., Ltd. (Beijing, China). Daidzin, daidzein, genistin, and genistein

were purchased from Shanghai Yuanye Bio-Technology Co., Ltd. (Shanghai, China). All other chemicals used in this study were analytical grade.

### 2.2. Strain Culture

The strain stored in a refrigerator at 4 °C was transferred to potato dextrose agar (PDA) medium under aseptic conditions. To ensure consistent culture conditions, the cultures were incubated at a constant temperature of 26 °C and 95% humidity in the dark for 7 d, until the mycelium covered the surface of the medium. The resulting mycelial block (0.7 cm$^2$) was inoculated into a liquid medium (comprising 20 g of glucose, 10 g of peptone, 3 g of $KH_2PO_4$, 1.5 g of $MgSO_4 \bullet 7H_2O$, 10 mg of vitamin $B_1$, and 1 L of distilled water). The liquid fermentation strain was cultured in the dark for 7 d at 26 °C and 160 rpm.

### 2.3. Solid-State Fermentation of Soybean Meal

The fungal SSF process according to the study by Asensio-Grau et al. [19] with slight modifications. To prepare the substrates for SSF, 5 g of soybean meal powder (SP) and 5 g of distilled water were added to fermentation bottles (100 mL), and then sterilized using an autoclave (model YXQ-LS-50A, Shanghai Boxun Industrial Co., Ltd., Shanghai, China) at 121 °C for 30 min. Then, 3 mL of culture suspension (3.2 mg/mL biomass, dry weight) was added to each sterilized bottle for SSF. The cultures were covered with tinfoil and incubated at 28 °C for 14 d in the dark. Upon completion of the incubation period, the fermented substrates were removed, transferred to a blast drying oven (GZX-9076MBE, Shanghai Boxun Industrial Co. Ltd., Shanghai, China), and dried at 60 °C for 24 h. To achieve a consistent particle size, the dried substrates were ground into powders and sifted through a 40-mesh sieve. The fermented soybean meal powders were separately obtained for each of the three mushrooms (*P. ostreatus*, *H. erinaceus*, and *F. velutipes*), corresponding to PFSP (*Pleurotus ostreatus* fermented soybean meal powder), HFSP (*Hericium erinaceus* fermented soybean meal powder), and FFSP (*Flammulina velutipes* fermented soybean meal powder). Then, the fermented powders were stored in PE self-sealing bags for further analysis and determination. SP without fermentation was sterilized under the same conditions as a control, corresponding to SSP (sterilized soybean meal powder). All groups were prepared in triplicate.

### 2.4. Macronutrients Content Analysis

Crude protein content was analyzed as accorded by AOAC 960.52 semi-micro Kjedahl method with a conversion factor of 6.25. Moisture content was determined following AOAC 925.10. The ash content was measured using direct ashing as per AOAC 930.05. The sugar content was analyzed as per AOAC 931.02.

### 2.5. Determination of Amino Acid Content

The amino acid content was determined according to the method of Wang et al. [20] with some modifications. An amount of 0.5 g of SPs was firstly hydrolyzed with isometric HCl (10 mL, 6 M) in a sealed hydrolysis tube. A small amount of phenol (3–4 drops) was added to the tube, and then the tube was frozen for 3–5 min. Subsequently, the acid hydrolysis tube was rinsed with nitrogen and the samples were hydrolyzed at 110 °C for 22 h. The resulting solution was subsequently diluted with distilled water, filtered through an aqueous phase filtration membrane (0.45 μm), dried under vacuum, and dissolved in 1 mL sodium citrate (pH 2.2). The resulting filtrate was analyzed for quantification using an amino acid automatic analyzer (LA8080, Hitachi High-Tech Corporation, Minato-ku, Tokyo, Japan).

### 2.6. Determination of Soybean Isoflavones

The soybean isoflavone content was determined as per AOAC 2001.10. The sample was pretreated by adding 0.10 g of the fermentation sample to 10 mL of an 80 vol% methanol solution, followed by 40 min of ultrasonication at 480 W. A suitable amount of the sample

solution was then centrifuged at 8000 rpm for 10 min in a high-speed centrifuge (Hitachi CR21N, Eppendorf Himac Technologies Co., Ltd., Ibaraki-ken, Japan). The supernatant was passed through a 0.45 μm filter membrane, and the soybean isoflavones were analyzed by high-performance liquid chromatography (LC-2030C 3D Plus, Shimadzu, Kyoto, Japan). The HPLC conditions were as follows: column: ZORBAX SB-C18 column (250 × 4.6 mm, Agilent, Santa Clara, CA, USA); mobile phase A: acetonitrile; mobile phase B: 2% acetic acid. 0–12.5 min, A: 10%; 12.5–17.5 min, A: 30%; 17.5–18.5 min, A: 40%; 18.5–26 min, A: 95%; 26–30 min: 95%; 26–30 min, A: 10%; flow rate 1 mL/min; wavelength 260 nm; injection volume 20 μL; column temperature 30 °C.

### 2.7. Determination of Total Polyphenol Content

Referring to the extraction method of Ali et al. [21], 0.10 g of the samples were individually weighed, and 10 mL of 80% methanol by volume was added. Sonicating for 60 min, and centrifuging at 8000 rpm and 25 °C for 10 min. The supernatant was then filtered through a 0.22 μm membrane to collect the filtrate, which was repeated twice. The total polyphenol content (TPC) was determined using the Folin–Ciocalteu method. A standard solution of 0.1 mg/mL gallic acid was prepared, and a sequence of standard solutions with mass concentrations of 0, 0.01, 0.02, 0.03, 0.04, and 0.05 mg/mL were also prepared. An amount of 0.5 mL of the series standard solution or sample solution was added, followed by 2.5 mL of Folin–Ciocalteu reagent. The mixture was shaken well and allowed to stand for 30 s before adding 2.0 mL of 7.5 g/100 mL sodium carbonate solution. The volume was fixed to 5.0 mL and allowed to react for 1 h at 760 nm in the dark. The absorbance value was determined by ultraviolet spectrophotometry (GENESYS 150, Thermo Fisher Scientific Inc, Waltham, MA, USA). TPC was expressed as milligram gallic acid equivalents per gram of sample (mg GAE/g).

### 2.8. Determination of Antioxidant Capacity

Weighed 0.10 g of the sample precisely and added 10 mL of 80% methanol. The samples were treated with ultrasonication at 480 W for 2 h and centrifuged for 10 min at 10,000 rpm and 25 °C in a high-speed centrifuge (Hitachi CR21N, Eppendorf Himac Technologies Co., Ltd., Ibaraki-ken, Japan). The resulting supernatant is used to determine DPPH radical scavenging capacity, hydroxyl radical scavenging capacity, and ABTS$^+$ radical scavenging capacity.

#### 2.8.1. Determination of DPPH Radical Scavenging Activity

With slight modifications to the method of Sánchez-García et al. [22], 2 mL of each sample solution was mixed with 2 mL of a 0.2 mM DPPH-95% ethanol solution (Ai), and the mixture was vortexed for 1 min and then incubated at room temperature for 30 min in the dark. The absorption value at 517 nm wavelength was measured. The blank control (A0) was measured using distilled water rather than the sample solution, and the background (Aj) was measured using 95% ethanol rather than DPPH-95% ethanol solution. The results were presented as DPPH radical scavenging activity according to Formula (1).

$$\text{DPPH radical scavenging activity } (\%) = \left(1 - \frac{\text{Ai} - \text{Aj}}{\text{A0}}\right) \times 100\% \tag{1}$$

In which Ai, A0, and Aj represent the sample, blank control, and background absorbances, respectively.

#### 2.8.2. Determination of Hydroxyl Radical Scavenging Activity

According to the method of Lee et al. [23] with few modifications, 1.0 mL of the sample was combined with 1.0 mL of a 9 mM solution of ferrous sulfate and 1.0 mL of a 9 mM solution of salicylic acid–ethanol. A further 1 mL of a 9 mM hydrogen peroxide solution was added, and the mixture was thoroughly blended. The resulting solution was incubated in a water bath maintained at 37 °C for 30 min. The sample absorbance (Ai) was detected at

510 nm using a spectrophotometer (GENESYS 150, Thermo Fisher Scientific Inc., Waltham, MA, USA). Additionally, the blank control (A0) was determined using 1.0 mL of distilled water instead of the sample, and the background (Aj) was determined using 1.0 mL of distilled water instead of hydrogen peroxide. The results were presented as hydroxyl radical scavenging activity with Formula (2).

$$\text{Hydroxyl radical scavenging activity } (\%) = \left(1 - \frac{Ai - Aj}{A0}\right) \times 100\% \tag{2}$$

In which Ai, A0, and Aj represent the sample, blank control, and background absorbances, respectively.

### 2.8.3. Determination of ABTS$^+$ Radical Scavenging Activity

The method was followed by Lee et al. [23] with slight modifications, a solution of 7.4 M ABTS$^+$ and 2.6 mM potassium persulfate were prepared and mixed in equal volumes. The mixture was kept in the dark for 12–16 h to allow the formation of mother liquor. This mother liquor was then diluted with 5 mM phosphate buffer solution (pH 7.4) until the absorbance at 734 nm reached 0.70 $\pm$ 0.02. To 1.8 mL of the ABTS$^+$ dilution solution, 0.2 mL of the sample solution was mixed (Ai). The mixtures were incubated in the dark for 10 min before measuring their absorbance values at 734 nm. The blank control (A0) was created by blending 1.8 mL of the ABTS$^+$ dilution solution with 0.2 mL of distilled water. The results were presented as ABTS$^+$ radical scavenging activity, calculated using Formula (3).

$$\text{ABTS}^+\text{radical scavenging activity } (\%) = \left(\frac{A0 - Ai}{A0}\right) \times 100\% \tag{3}$$

In which Ai, A0, and Aj represent the sample, blank control, and background absorbances, respectively.

### 2.9. Color

The color of fermented SP was measured using a colorimeter (CQX3448, Hunter Associates Laboratory, Inc., VA, USA) calibrated at room temperature. Different samples of the powders were evaluated, and color difference data were obtained by randomly taking three measurements at different locations on the surface of the powders. Each measurement was performed three times, with the mean value obtained. L (0 = black, 100 = white), a (+ value = red, − value = green) and b (+ value = yellow, − value = blue) were recorded. The color difference ($\Delta$E) was calculated using Equation (4).

$$\Delta E = \sqrt{\Delta L^2 + \Delta a^2 + \Delta b^2} \tag{4}$$

### 2.10. Particle Size

The particle size distributions of soybean meal powders and fermented soybean meal powders were assessed using a dynamic light scattering instrument (Zetasizer NanoZS90, Malvern Instruments, Worcestershire, UK). The refractive index used for the dispersed phase was established at 1.45 and the refractive index of water (continuous phase) was established at 1.33. The results were reported as the average of the mean particle (d3,2).

### 2.11. Scanning Electron Microscope (SEM)

The microstructure of the samples was evaluated by scanning electron microscopy (SEM). The sample was placed in an ion sputtering apparatus, and then coated with a thin layer of gold before imaging and observed under the SEM (Zeiss Gemini 500, Carl Zeiss AG, Germany) at different magnifications (500$\times$ and 1000$\times$) with an accelerating voltage of 5 kV.

### 2.12. Statistical Analysis

Sample preparations were conducted in triplicate. All the analyses (except SEM) were repeated in triplicate. The data were presented as mean ± standard deviation (SD). The graphs were designed using Origin 2021 (Origin Lab Co., Ltd., Northampton, MA, USA). The results were statistically analyzed using one-way ANOVA (Analysis of variance) according to Duncan's test with SPSS Statistics 19 (SPSS, Inc., Chicago, IL, USA). Statistical significance was defined as a *p*-value less than 0.05.

## 3. Results

### 3.1. Macronutrients Content

As shown in Table 1, a comparison between unsterilized soybean meal powders (SP) and sterilized soybean meal powders (SSP) revealed a significant decrease in the crude protein content of the latter. This may be due to the structural changes caused by autoclaving proteins, exposing hydrophobic amino acids, and forming new covalent and non-covalent bonds [24]. These changes promote protein aggregation, resulting in decreased protein solubility [25]. In contrast, fermented SPs have been found to contain significantly higher levels of crude protein than SSP, consistent with previous research [26]. This increase can be attributed to the partial hydrolysis of nutrients by microbial enzymes (such as protease) during fermentation, and the conversion of some carbohydrates to protein, peptides, or even free amino acids [19]. Additionally, protein content can increase due to cell growth and an increase in protein biomass [27]. As the Kjeldahl method does not differentiate between different sources of nitrogen, the increase in assay results may come from various nitrogen-containing components. Nitrogenous components in biomass are protein, free amino acids, nucleic acids, ammonia, nitrates, amines, urea, and others. Mycelium also contains high levels of RNA [26,28]. Amino acid analysis is required for better validation of the results.

**Table 1.** Nutritional components of different soybean meal powders.

| Groups | Crude Protein (%) | Sugars (%) | Moisture (%) | Ashes (%) |
|--------|-------------------|------------|--------------|-----------|
| SP | 48.39 ± 3.46 [b] | 27.15 ± 2.07 [a] | 11.41 ± 0.42 [a] | 6.39 ± 0.28 [a] |
| SSP | 38.42 ± 3.29 [c] | 21.92 ± 1.86 [b] | 7.38 ± 0.35 [c] | 5.87 ± 0.31 [b] |
| PFSP | 52.98 ± 3.87 [a] | 17.98 ± 1.73 [c] | 7.30 ± 0.38 [c] | 5.44 ± 0.24 [c] |
| HFSP | 51.95 ± 3.14 [a] | 20.67 ± 1.47 [b] | 7.55 ± 0.29 [b] | 5.74 ± 0.43 [b] |
| FFSP | 51.18 ± 3.55 [a] | 18.86 ± 1.62 [c] | 7.46 ± 0.31 [b] | 5.49 ± 0.37 [c] |

SP: soybean meal powder; SSP: sterilized soybean meal powder; PFSP: *Pleurotus ostreatus* fermented soybean meal powder; HFSP: *Hericium erinaceus* fermented soybean meal powder; FFSP: *Flammulina velutipes* fermented soybean meal powder. Data are mean ± SD of triplicates, values in the same column with different letters (a–c) present significant difference (*p* < 0.05).

The protein content of PFSP was slightly higher than other fermented soybean meal powders (HFSP and FFSP). This result may be due to the greater capacity of *Pleurotus* species to adapt to this substrate for biotransformation using carbon sources. The acidophilic nature of *Pleurotus* species, as well as their ability to lower pH through the release of organic acids, have been shown to prevent nitrogen losses due to ammonia volatilization [16].

SSP was found to have significantly lower total sugar content than SP. This can be attributed to the leaching of soluble sugars during the autoclaving process, as previously reported [19]. The dietary fiber in soybean meal is mainly composed of cellulose, hemicellulose, and pectin, which are important components of the plant cell wall. The thermal and stress effects of the autoclaving process result in cell wall breakdown, partial degradation of high molecular weight polysaccharides, and an increase in soluble dietary fiber [24]. This corresponds to the change in total sugar content. The total sugar content of fermented SPs was significantly lower than that of SSP. This may be due to the growth and development of edible mushroom mycelium. Microbial enzymes such as amylase, cellulase, hemicellulose, and *β*-glycosidase hydrolyze carbohydrates into monosaccharides, which are utilized by microorganisms as a source of energy [19]. The total sugar content of PFSP was slightly less than that of HFSP and FFSP. This discrepancy may be due to the different growth rates of

the different fungal strains. As previously reported [27], *P. ostreatus* has a superior capacity for utilizing carbohydrates. This explains the lower total sugar content of PFSP compared to other fermented soybean meal powders (HFSP and FFSP). In general, the way mushroom mycelium absorbs nutrients depends on species, substrates, and enzymes [16,29], which could explain the differences in nutrient utilization between the three edible mushrooms.

*3.2. Amino Acid Content*

The heat map summarized the overall changes in amino acid composition for different SPs (Figure 1). The changes in the amino acid content of different SPs are shown in Table S1. Compared to SP, the total free amino acid concentration of SSP increased slightly. The autoclaving process did not cause significant amino acid loss. However, basic amino acids such as Lys and Arg decreased slightly, which could be related to the extent of the Maillard reaction [30]. These amino acids may also be destabilized by the acidic conditions related to fermentation [16]. Total free amino acid concentration increased significantly during fermentation. This change mainly contributed to an increase in the concentrations of Asp, Thr, Glu, and Ala. These free glucogenic amino acids (alanine, aspartic acid, and glutamic acid) can be involved in gluconeogenesis [29], which may partly explain the increase in total sugars. It is also possible that the rate of protein hydrolysis exceeded the rate of amino acid catabolism, resulting in an increase in free amino acids after fermentation. Notably, the concentration of glutamate and aspartic acid increases significantly after fermentation (PFSP), by 23.29% and 23.16%, respectively. This may be due to their presence as the main amino acids in edible mushrooms. Glutamic acid and aspartic acid are natural umami amino acids that can be used to enhance the flavor of low-salt foods. Moreover, the increase in amino acids during fermentation is attributed to the hydrolysis of proteins into free amino acids by microbial enzymes [30]. In contrast, the amino acid content of PFSP increased more obviously. During the fermentation process, the synthesis and utilization of free amino acids by the mushrooms would cause changes in amino acid content and composition [31]. According to the study by Zhao, et al. [32], *P. ostreatus* and other edible mushrooms can use okara protein as a nitrogen source to generate more free amino acids.

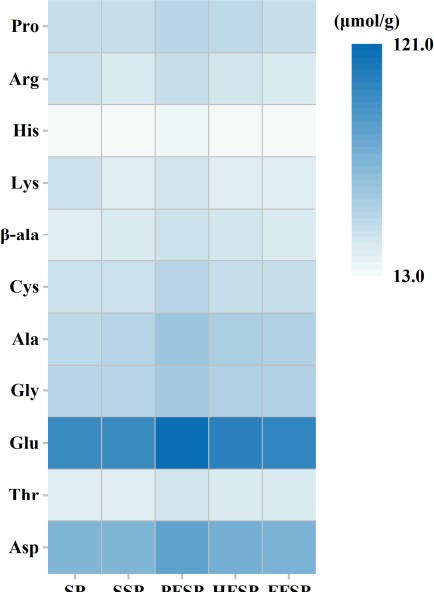

**Figure 1.** Heat map of amino acid composition in different soybean meal powders. SP: soybean meal powder; SSP: sterilized soybean meal powder; PFSP: *Pleurotus ostreatus* fermented soybean meal powder; HFSP: *Hericium erinaceus* fermented soybean meal powder; FFSP: *Flammulina velutipes* fermented soybean meal powder.

Some amino acids and peptides may also contribute to the antioxidant capacity of fermented soybean meal (Section 3.5). Moreover, glutamic acid and aspartic acid can impart freshness to the samples. Overall, the alteration of amino acid content by fungal SSF affects the nutritional value and sensory quality of soybean meal.

### 3.3. Isoflavone Content

The liquid chromatogram revealed the alteration of isoflavones in different SPs (Figure 2). The changes in isoflavone content of different SPs are shown in Table S2. As soybean meal undergoes thermal processing, the content of aglycones (genistein, daidzein, etc.), decreases while the content of glycosides increases (genistin, daidzin, etc.). Studies have reported that the isoflavone aglycone content decreased significantly after wet heating, which is related to the heating time [33]. This may be due to protein-bound isoflavones being released during autoclaving. The deesterification of acetylglycoside isoflavones could explain an increase in glycosides under wet heating [33]. However, compared to the autoclaving process, fermentation has a more significant impact on isoflavone content. Fungal SSF dramatically enhanced the aglycone content while reducing the glycoside content. Soybean isoflavone glycosides increased by 100.54–225.92% in daidzein and 148.05–333.60% in genistein (Table S2). This is consistent with the results of Xu, et al. [9], who studied the significant increase in the content of isoflavone aglycones in fermented soy foods. These data indicated that the isoflavone glycosides were effectively converted into their corresponding aglycones. $\beta$-glycosidase is secreted by microorganisms and plays a key role in the accumulation of nutrients and the conversion of components [34]. It can be converted from the glycosides form (genistin, daidzin, etc.), to the aglycones form (genistein, daidzein, etc.), the latter being more bioavailable and more easily digested and absorbed by the human body [20]. During fermentation, $\beta$-glycosidase secreted by edible mushrooms catalyzes the conversion of glycosides into aglycones. The PFSP contained a higher volume of aglycones than HFSP and FFSP. *P. ostreatus*, as a white rot fungus, has a high activity of endoglucanase, exoglucanase, and $\beta$-glycosidase [35]. These enzymes can hydrolyze isoflavone glycosides to isoflavone aglycones. In summary, both autoclaving and fungal SSF processes can convert isoflavones into glycosides, but fermentation has a greater impact on the overall isoflavone content.

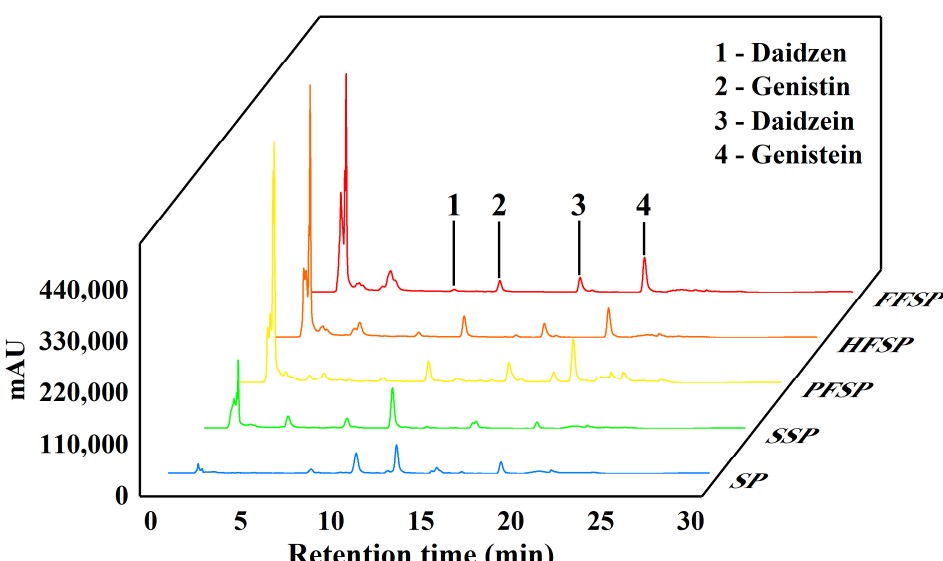

**Figure 2.** Liquid chromatogram of soybean isoflavones in different soybean meal powders. SP: soybean meal powder; SSP: sterilized soybean meal powder; PFSP: *Pleurotus ostreatus* fermented soybean meal powder; HFSP: *Hericium erinaceus* fermented soybean meal powder; FFSP: *Flammulina velutipes* fermented soybean meal powder.

### 3.4. Total Polyphenol Content

The changes in total polyphenol content (TPC) of different SPs are shown in Figure 3a. High-temperature sterilization is a necessary step in SSF to prevent stray bacteria from contaminating the fermentation substrate. Compared to the SP, the TPC was slightly reduced after autoclaving due to the destruction of polyphenols caused by this treatment [26]. This result is consistent with previous studies, which have shown that autoclaving treatment can reduce the TPC in buckwheat [36]. The TPC of SP and SSP was $1.255 \pm 0.077$ mg GAE/g and $1.157 \pm 0.071$ mg GAE/g, respectively. However, the TPC of PFSP, HFSP, and FFSP was $4.215 \pm 0.226$ mg GAE/g, $3.183 \pm 0.098$ mg GAE/g, and $2.751 \pm 0.067$ mg GAE/g, respectively. Previous studies have noted that SSF treatment of mung beans and soybeans can improve the TPC from $11.62 \pm 0.02$ mg GAE/g and $4.59 \pm 0.02$ mg GAE/g to $38.39 \pm 1.47$ mg GAE/g and $22.56 \pm 0.31$ mg GAE/g, respectively [21]. Fermentation of beans with *Bacillus subtilis* was reported to increase the TPC from $15.89 \pm 0.56$ mg GAE/g to $35.93 \pm 0.69$ mg GAE/g [37]. SSF with different edible mushroom mycelia has been shown to significantly affect total polyphenol content. Edible mushroom mycelium uses substrate nutrients for growth and development, generating a variety of enzymes such as $\beta$-glycosidases and cellulases. These enzymes break down macromolecules and liberate individual phenolic compounds from the substrate, enhancing antioxidant capacity [34]. Furthermore, the antioxidant activity of SP was enhanced by the release of lipophilic aglycones from isoflavone glycosides through the catalytic action of $\beta$-glucosidase [34]. The TPC in the PFSP increased more significantly. This is likely the result of conjugated polyphenol depolymerization or hydrolysis, as well as phenylalanine deamination [16].

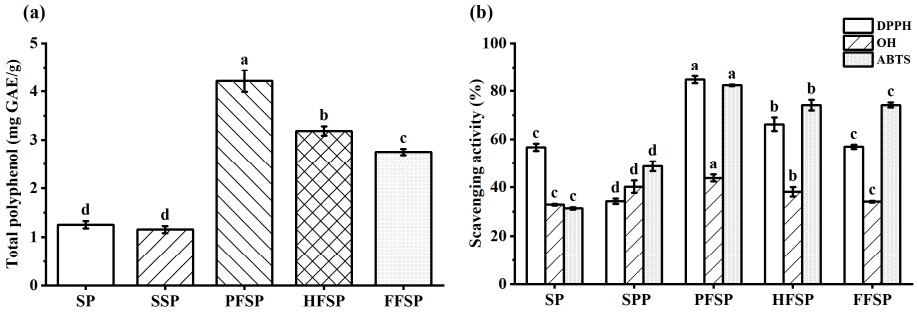

**Figure 3.** Changes in total polyphenol content (TPC) and antioxidant activity during the fermentation of different soybean meal powders. (**a**) Total polyphenol content; (**b**) scavenging activity. SP: soybean meal powder; SSP: sterilized soybean meal powder; PFSP: *Pleurotus ostreatus* fermented soybean meal powder; HFSP: *Hericium erinaceus* fermented soybean meal powder; FFSP: *Flammulina velutipes* fermented soybean meal powder. Data are mean $\pm$ SD of triplicates, values with different letters (a–d) present significant difference ($p < 0.05$).

### 3.5. Antioxidant Capacity

SSF can cause changes in the substrate that result in bioactive secondary metabolites, thereby enhancing their antioxidant activity. Each assay for measuring antioxidant capacity has a different theoretical principle, and a single assay cannot accurately reflect the antioxidant potential of fermented samples. Therefore, in this study, three different assays (DPPH, •OH, ABTS$^+$) are explored to analyze unfermented and fermented samples (Figure 3b). The results showed that SSF considerably improved the antioxidant capacity of soybean meal.

The comparisons of the DPPH free radical scavenging activities of different SPs are shown in Figure 3b. The DPPH radical scavenging activities of the SP and SSP were found to be $56.70 \pm 1.51\%$ and $34.24 \pm 1.09\%$, respectively. On the contrary, the DPPH radical scavenging activity of the fermented soybean meal powder with the three edible mushrooms strains was found to be significantly higher, with values of $84.73 \pm 1.45\%$, $66.24 \pm 2.79\%$, and $56.95 \pm 0.86\%$ for *P. ostreatus*, *H. erinaceus*, and *F. velutipes*, respectively. This can

be partly accounted for by the presence of organic compounds such as aromatic amines, amino acids, and peptides. These are released during SSF and react with the DPPH radical. During SSF, the edible mushroom mycelium generated hydrolytic enzymes that catalyzed the release of conjugated phenolic compounds, thereby increasing antioxidant activity. The DPPH radical scavenging activity of lentils was reported to increase significantly after fermentation with *Aspergillus awamori* [38].

The comparisons of the hydroxyl radical scavenging activities of different SPs are shown in Figure 3b. The hydroxyl radical scavenging activities of SP and SSP were $40.25 \pm 2.51\%$ and $32.84 \pm 0.35\%$, respectively. However, the hydroxyl radical scavenging activity of SP after fermentation by the three strains, *P. ostreatus*, *H. erinaceus*, and *F. velutipes*, was $43.81 \pm 1.48\%$, $38.13 \pm 1.98\%$, and $34.13 \pm 0.34\%$, respectively. The presence of organic acids in the fermentation substrate, such as free hydroxyl phenols and soybean polyphenols, can provide active hydrogen to terminate the free radical chain reaction, explaining the increase in hydroxyl radical scavenging activity after fermentation [38]. As illustrated in Figure 3b, the hydroxyl radical scavenging capacity of all fermented SPs was significantly lower than that of other free radicals. Our results revealed that the hydroxyl radical scavenging effects were detected at lower values than the DPPH and ABTS$^+$ radicals. Some studies have demonstrated that lentil fermentation with *Aspergillus awamori* resulted in higher hydroxyl radical scavenging activity, showing a maximum value of 43.1%, an increase of 89% compared to unfermented samples [38].

Comparisons of ABTS$^+$ radical scavenging activities of different SPs are shown in Figure 3b. The results show that the ABTS$^+$ radical scavenging activities of SP and SSP before fermentation were $48.75 \pm 2.08\%$ and $31.34 \pm 0.47\%$, respectively. The ABTS$^+$ radical scavenging activities of SP after fermentation by the three strains, *P. ostreatus*, *H. erinaceus*, and *F. velutipes*, were $82.34 \pm 0.30\%$, $74.17 \pm 2.19\%$, and $74.22 \pm 1.08\%$, respectively. These results demonstrate that the microbial fermentation process significantly affected the ABTS$^+$ radical scavenging activity of the substrate. In particular, it was observed that the soybean meal fermented with *P. ostreatus* had the highest ABTS$^+$ radical scavenging activity, with an increase of 33.59% and 51% compared to SP and SSP, respectively. This can be explained by the superior ability of *P. ostreatus* to biotransform isoflavone glycosides into aglycones isoflavone. These have a higher antioxidant capacity. Similar results were discovered in the research of Sánchez-García et al. [22], the result showing that the ABTS$^+$ radical scavenging activity of fermented lentils and quinoa was significantly increased.

Overall, the antioxidant scavenging capacity of the three free radicals in solid fermentation of edible mushroom mycelia was ABTS > DPPH > hydroxyl radical in order. The observed increase in antioxidant capacity in fermented soybean meal can be attributed to several factors in the complex fermentation process. The main sources of antioxidant capacity in soybean meal are isoflavones, flavonoids, polyphenol compounds, and soy saponins, particularly catechol or ortho-benzotriol groups [39]. These compounds have stronger antioxidant activity than other groups, such as hydroxyl or ketone groups, which contribute significantly to their antioxidant capacity by providing individual electrons [39]. The antioxidant capacity increased slightly after heating, possibly due to increased polyphenolic compound solubility and thermal degradation of isoflavone glycosides [29]. Soybean isoflavone aglycones had higher antioxidant capacity than their glycosides. These aglycones have a hydroxyl group that reacts with free radicals, acting as an oxygen donor and terminating the chain reaction of free radicals [34]. Additionally, the presence of hydrophobic and aromatic amino acids in soybean meal, such as histidine and phenylalanine, may contribute to its antioxidant properties [40]. Furthermore, the hydrolysis of soybean meal proteins during fermentation results in the release of small peptides, which have been shown to be positively correlated with increased antioxidant activity [41]. In conclusion, SSF of soybean meal with edible mushroom mycelia significantly increases its antioxidant capacity and bioavailability, making it a potential novel food material or nutritional supplement. This is important for the prevention of free radical-related diseases such as neurodegenerative diseases, cancer, cardiovascular diseases, and others [4].

### 3.6. Color

Fermentation affected the optical properties as shown in Table 2, with significant changes in lightness (L*), redness (+a*), and yellowness (+b*) detected. The autoclaving process induced a significant lightness (L*) decrease in the prepared SP, and an increase in the a* value from $0.02 \pm 0.20$ to $10.45 \pm 0.74$, indicating a change towards a red color. The color of the fermented soybean powder was similar to that of the SSP. However, compared to the other fermented soybean meal powders (HFSP and FFSP), the color of PFSP was observed to be yellowish. This color change can be attributed to the degradation of pigments and fungal SSF [42]. It should be noted that thermal drying and milling are processes that alter the physical properties of the samples, which can lead to the darkening of fermented soybean meal powders.

**Table 2.** Color changes in different soybean meal powders.

| Color Coordinates | L* | a* | b* | ΔE |
|---|---|---|---|---|
| SP | 84.33 ± 2.27 [a] | 0.02 ± 0.20 [e] | 14.32 ± 0.40 [c] | — |
| SSP | 50.98 ± 1.54 [c] | 10.45 ± 0.74 [b] | 16.83 ± 0.92 [b] | 35.06 ± 1.18 [a] |
| PFSP | 57.68 ± 0.45 [b] | 11.36 ± 0.35 [a] | 21.08 ± 0.60 [a] | 29.75 ± 0.36 [b] |
| HFSP | 51.72 ± 1.01 [c] | 8.45 ± 0.01 [c] | 15.13 ± 0.67 [c] | 33.70 ± 0.96 [a] |
| FFSP | 57.85 ± 1.03 [b] | 7.37 ± 0.55 [d] | 15.17 ± 0.66 [c] | 27.51 ± 0.87 [c] |

SP: soybean meal powder; SSP: sterilized soybean meal powder; PFSP: *Pleurotus ostreatus* fermented soybean meal powder; HFSP: *Hericium erinaceus* fermented soybean meal powder; FFSP: *Flammulina velutipes* fermented soybean meal powder. Data are mean ± SD of triplicates, values in the same column with different letters (a–e) present significant difference ($p < 0.05$).

### 3.7. Particle Size

Fermentation resulted in slight changes in the particle size distribution of the samples. The mean size (d3,2) of the fermented soybean meal powders were ($1226.67 \pm 164.85$), ($966.00 \pm 26.51$), ($1271.33 \pm 41.02$) for PFSP, HFSP, and FFSP, respectively, all of which had smaller particles than the unfermented soybean meal powder, d3,2 ($1461.00 \pm 62.22$). This variation may be related to fungal degradation or consumption of the substrate [19]. The decrease in particle size of fermented SP can be attributed to the increased degree of substrate fragmentation, which leads to the exposure of more polar groups and easier release of nutrients from the substrate. Edible mushroom mycelia can utilize this to enhance the accessibility, digestibility, and bioavailability of enzyme substrates.

### 3.8. SEM

The scanning electron microscope images illustrate the morphological changes in different SPs. Figure 4a shows unfermented soybean meal (SP) with different starch granules and protein matrix structures with well-defined boundaries. The starch granules are irregularly oval and round with a smooth surface. The surrounding protein matrix or cellulose forms irregular clusters. Figure 4b shows that after autoclaving, the surface structure of SP becomes irregular and rough. Protein aggregation and protein folding were also observed. This was due to the interaction between the protein granules caused by the high temperature and pressure. Such granules with an irregular surface can improve the use of this biomass as a SSF substrate and facilitate the growth of fungal colonization. The surface of fermented SP showed a porous network structure with a distorted paste-like surface, wrinkles, and aggregated surfaces (Figure 4d–f). Compared to unfermented SP, microbial fermentation disrupted the microstructure of the soybean meal surface, indicating that microorganisms decomposed and used proteins and other nutrients in the soybean meal [20]. Furthermore, small fragmented lumpy structures were found in all fermented SP, and the clear boundaries between the protein fragments had disappeared. This could be due to the formation of more protein aggregates during fermentation in the presence of proteases, and the loose porous structure of the samples can improve their adsorption properties [20]. Different edible fungal fermentations of soybean meal have their own morphological characteristics that are influenced by substrates and enzymes. Soybean

meal substrates have the potential to support the growth and development of different mushroom mycelia.

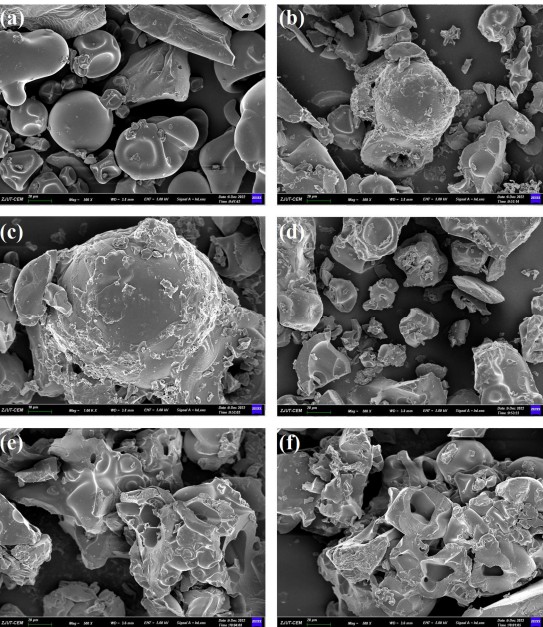

**Figure 4.** Scanning electron microscopy (SEM) images of different soybean meal powders. (**a**) SP 500×; (**b**) SSP 500×; (**c**) SSP 1000×; (**d**) PFSP 500×; (**e**) HFSP 500×; (**f**) FFSP 500×. SP: soybean meal powder; SSP: sterilized soybean meal powder; PFSP: *Pleurotus ostreatus* fermented soybean meal powder; HFSP: *Hericium erinaceus* fermented soybean meal powder; FFSP: *Flammulina velutipes* fermented soybean meal powder. The magnification factors were 500× and 1000×.

## 4. Conclusions

This study revealed the positive influence of fungal SSF on the nutritional composition, physicochemical properties, and bioaccessibility of soybean meal. Compared with unfermented soybean meal, fermented soybean meal powder demonstrated an enhanced protein profile, antioxidant capacity (as evaluated by DPPH and ABTS⁺ assays), and processing potential. In addition, the fungal SSF can convert glycosides into aglycones isoflavones, which are more easily absorbed by the body. The SEM observations revealed a looser and more porous microstructure, which facilitated the digestion of the samples. Furthermore, *P. ostreatus* has displayed a greater efficiency in utilizing soybean meal for fermentation than the other two edible mushrooms (*H. erinaceus* and *F. velutipes*), resulting in higher nutritional value and improved physicochemical properties. Fungal SSF presents a potential way to enhance the properties of SP by the bioconversion of agricultural by-products into value-added products. These fermented soybean meal powders have the potential to serve as a substitute for plant-derived proteins, such as those used in emerging plant-based meat products, thus yielding significant economic value and social benefits. Therefore, more research is needed on the potential health benefits of multi-strain co-fermentation within this context, with follow-up experiments focusing on the changes in the active compounds of edible mushrooms resulting from SSF.

**Supplementary Materials:** The following supporting information can be downloaded at: https://www.mdpi.com/article/10.3390/fermentation9040322/s1, Table S1: The changes in amino acid content of different soybean meal powders; Table S2: The changes in isoflavone content of different soybean meal powders.

**Author Contributions:** Conceptualization, J.W.; data curation, Z.H.; formal analysis, M.C.; funding acquisition, J.W. and K.Y.; investigation, Q.J. and Z.H.; methodology, Q.J. and Z.H.; project administration, K.Y. and P.S.; resources, Y.W.; software, Z.H.; supervision, J.W., M.C. and P.S.; visualization, Q.J.; writing—original draft, Q.J.; writing—review and editing, J.W. and H.R. All authors have read and agreed to the published version of the manuscript.

**Funding:** This research was funded by the Zhejiang Provincial Natural Science Foundation (grant number LQ23C200013) and Science and Technology General Project of Longyou (grant number JHXM2022004).

**Institutional Review Board Statement:** Not applicable.

**Informed Consent Statement:** Not applicable.

**Data Availability Statement:** Data reported in this study are contained within the article. The underlying raw data are available on request from the corresponding author.

**Conflicts of Interest:** The funders had no role in the design of the study; in the collection, analyses, or interpretation of data; in the writing of the manuscript; or in the decision to publish the results.

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
