# Peer review of "Solid-State Fermentation of Soybean Meal with Edible Mushroom Mycelium to Improve Its Nutritional, Antioxidant Capacities and Physicochemical Properties"

_fermentation, doi:10.3390/fermentation9040322_

Round 1

Reviewer 1 Report

In the manuscript the authors examine the effect of solid-state-fermentation with 3 different edible fungi on proximate and antioxidant composition of soybean meal.

The authors presented this theme based on 40 cited articles, what seem to be adequate to the specific field of the manuscript.

The presented data are interesting, however I have some comments.

1)       Table S1 in supplementary material: the quantification of amino acids is indicated in table without any indication of repetiotion number, means and standatd deviation and any statistical indication.

Please provide provide SD and at least Anova

2)       Table S2 in supplementary material: The changes in isoflavone content of different soybean meal powders.

Can you provide an indication of unit of measure? How many repetitions? And also in this case please add SD and statistical data.

3)       In figure 3 the data are expressed as mg/g or mg GAE/g?

Please standardize the nomenclature of phenols content (mg/ or mg GAE/g)  throughout the manuscript (in paragraph 3.4).

4)       In the conclusion the authors state that P. ostreatus used soybean meal for fermentation  is more efficiently than the other two edible fungi. So they can definitely suggest the use of this fungus for soybean fermentation?

Please better describe and discuss the unique feature of this research.

Reviewer 2 Report

The manuscript entitled “Solid state fermentation of soybean meal with edible fungi to improve its nutritional, antioxidant capacities and physicochemical properties” aimed to use solid state fermentation to improve the bioactive activity of soybean meal. This manuscript is well-written and can be considered for publication after minor revision.

1.     It is suggested that the abstract can be more informative.

2.     P. ostreatus, F. velutipes, and H. erinaceus are mushrooms. Fungi are a diverse group of organisms that include not only mushrooms but also molds, yeasts, and other types of fungi. It is suggested to use mushrooms instead of fungi in the manuscript.

3.     As shown in Table 1, mushroom fermentation seemed to increase the crude protein, as compared with the SP group. Any possible explanations?

Reviewer 3 Report

Is the result written together with the discussion?

Why choose these three ediblb fungi?What is the proportion of them added in soybean meal and how are they determined?

36.

What’s mean of bio-accessibility?

45~79.

The paragraph could benefit from breaking it into shorter, more concise sentences to improve readability. Some sentences can be merged, and the use of transition words could be increased to create better cohesion between ideas. Additionally, the paragraph could benefit from using more active voice to improve clarity and eliminate passive phrasing.

98~105.

Can provide more context or explanation of the experimental setup and its significance. For example, it would be helpful to include more information on why a constant temperature of 26 °C and 95% humidity in the dark was chosen as the ideal culture conditions, and why a 7% inoculum was used. Additionally, it may be useful to provide more information on the purpose of the experiment and what the researchers are trying to achieve through this fermentation process.

107.

 it is not clear what modifications were made to the process. This could be elaborated on to provide more detail.

110.

 Could be revised to read "using an autoclave (model YXQ-LS-50A, Shanghai Boxun Industrial Co., Ltd, Shanghai, China) at 121°C for 30 minutes."

112.

could be revised to clarify that the cultures were incubated in the dark.

116

It would be helpful to know why the powders were stored in PE self-sealing bags in sentence

117.

could be revised to specify that the fermented soybean meal was obtained separately for each of the three fungi (P. ostreatus, H. erinaceus, and F. velutipes), rather than all together.

237~255

The paragraph could be improved by breaking it down into smaller, more concise sentences. Additionally, it would be helpful to provide more context on what the abbreviations and acronyms stand for, such as SP and SSP. The paragraph could also benefit from more explanation and elaboration on the results and findings, particularly in regards to the significance of the differences in protein content between the different samples.

279~300

The paragraph could benefit from more contextual information. For example, it would be useful to explain why changes in amino acid content are important and how they affect the nutritional value of soybean meals.

291

The sentence "As we all know" is unnecessary and should be removed.

387~400

One potential improvement to the paragraph could be to provide more context for the significance of the findings.

492,

consider adding "of soybean meal" after "nutritional composition" for clarity.

495,

consider specifying which type of antioxidant activity was measured

Round 2

Reviewer 3 Report

The conclusion can as concise as possible.

Author Response

Thank you for the suggestion. The conclusion has been revised to be more concise.